# Tularemia as a Mosquito-Borne Disease

**DOI:** 10.3390/microorganisms9010026

**Published:** 2020-12-23

**Authors:** Zakaria Abdellahoum, Max Maurin, Idir Bitam

**Affiliations:** 1Laboratoire Biodiversité et Environnement: Interaction Génome, Faculté des Sciences Biologique, Université des Sciences et de la Technologie Houari Boumediene, Alger 16111, Algeria; zabdellahoum@gmail.com; 2Centre National de Référence des Francisella, Institut de Biologie et de Pathologie, Centre Hospitalier Universitaire Grenoble Alpes, 38043 Grenoble, France; 3Centre National de la Recherche Scientifique, TIMC-IMAG, UMR5525, Université Grenoble Alpes, 38400 Saint Martin d’Heres, France; 4Ecole Supérieure des Sciences de l’Aliment et des Industries Alimentaires, Alger 16004, Algeria

**Keywords:** tularemia, *Francisella tularensis*, *Francisella tularensis* subsp. *holarctica*, Scandinavia, arthropod vectors, mosquitoes

## Abstract

*Francisella tularensis* (Ft) is the etiological agent of tularemia, a disease known for over 100 years in the northern hemisphere. Ft includes four subspecies, of which two are the etiologic agents of tularemia: Ft subsp. *tularensis* (Ftt) and Ft subsp. *holarctica* (Fth), mainly distributed in North America and the whole northern hemisphere, respectively. Several routes of human infection with these bacteria exist, notably through bites of Ixodidae ticks. However, mosquitoes represent the main vectors of Fth in Scandinavia, where large tularemia outbreaks have occurred, usually during the warm season. The mechanisms making mosquitoes vectors of Fth are still unclear. This review covers the inventory of research work and epidemiological data linking tularemia to mosquitoes in Scandinavia and highlights the gaps in understanding mosquitoes and Ft interactions.

## 1. *Francisella tularensis* and Tularemia

### 1.1. Historical Background 

Ft (originally named *Bacterium tularense*) was first isolated in 1911 by McCoy and Chapin in Tulare County (CA, USA), during an investigation of a epizootic pseudoplague in ground squirrels [1,2]. This bacterium was isolated from humans in 1912 by Werry and Lamb from a patient suffering from deer fly fever [3]. In 1921, Francis proposed the name tularemia for the disease caused by Ft [4]. In 1924, Parker and Spencer isolated the bacterium from the tick species *Dermacentor andersoni* and demonstrated this arthropod species’ role as an Ft vector [5]. The genus name *Francisella* and species name *tularensis* were proposed in 1947 to honor Edward Francis [6].

### 1.2. Francisella tularensis 

Ft is a Gram-negative, coccobacillus shaped, facultative intracellular bacteria. This species includes four subspecies distinguishable by their geographical distribution [7,8]. The most virulent subspecies are Ftt (type A) and Fth (type B), representing the etiological agents of tularemia [9,10]. The third subspecies, *mediasiatica,* has never been associated with human infections [11,12]. Finally, Ft subsp. *novicida* (Ftn) living in aquatic biotopes generally affects immunocompromised people and is used as a laboratory model due to its low virulence [13,14].

Ft was classified by the CDC in 2002 as a biological weapon category A [15] because of its high virulence and the possibility of inducing fatal pneumonia by aerosol diffusion [16]. Indeed, 10 CFU of this bacterium can cause infection in humans [15]. No tularemia vaccine is currently authorized for human use. In the past, tularemia vaccines were mainly developed to protect human populations from Ft as a biological weapon. The live vaccine strain (LVS) of Ft has been extensively used for this purpose but then abandoned because of significant side effects and low efficacy in preventing type A tularemia [10].

Ft is a zoonotic agent with a large animal reservoir (mammals, fish, amphibians, birds, reptiles). Lagomorphs (wild hares and rabbits) and small rodents (mice, voles, gerbils, lemmings, coypu, etc.) are the primary sources of human infections [3,10]. Humans can be infected through different ways, such as direct contact with infected animals, inhalation of contaminated dust, ingestion of contaminated food, contact with or ingestion of contaminated water, and arthropod bites (mainly the ticks Ixodidae) [17].

### 1.3. Tularemia Geographical Distribution 

Tularemia is mainly distributed in the Northern Hemisphere of the globe [18]. In Europe, Scandinavia (Sweden and Finland) is a primary endemic area, followed by the Balkans, particularly Kosovo [19,20]. Hungary and the European part of Turkey (Thrace) also record high incidences, followed by Slovakia, the Czech Republic, Serbia, Bulgaria, Norway, Germany, Spain, Poland, Georgia, and France [6,20,21,22]. In Asia, Russia, China, Japan, Kazakhstan, and Turkmenistan are endemic areas of tularemia. Northern America is also an endemic area for tularemia (mainly Ftt), with Canada and the United States recording several human infections every year and occasional outbreaks. In the United States, most human tularemia cases are reported in central states, Arkansas, Oklahoma, South Dakota, Kansas, and Missouri [20]. The subspecies *holarctica* has recently been detected in southern Australia, causing human infections after bites from possums [23]. Tularemia is classically absent in the United Kingdom, Iceland, Africa, South America, and Antarctica [20].

### 1.4. Tularemia, the Disease

Tularemia usually manifests in humans by a flu-like syndrome occurring on average 3 to 5 days after infection, with a maximum incubation period of two weeks [6,10,24]. Then, the disease classically progresses to one of the six clinical forms (sometimes combined), depending on the route of contamination. The ulceroglandular form manifests by a skin lesion at the inoculation site of bacteria and regional lymphadenopathy. The glandular form manifests by regional lymphadenopathy without skin lesions. The oculoglandular form corresponds to the bacterium inoculation through the conjunctiva. It corresponds to conjunctivitis with satellite lymphadenopathy. The oropharyngeal form occurs after oral contamination. It corresponds to pharyngitis with submandibular or cervical lymphadenopathy. The pneumonic form is triggered by the bacteria’s inhalation and corresponds to acute, subacute, or even chronic pneumonia. Finally, the typhoidal form manifests by severe sepsis usually associated with neurological signs (confusion) and Ft bacteremia. These two last systemic infections are the most severe forms of tularemia [10].

## 2. Review Objectives

The role of hematophagous arthropods as vectors of Ft is well characterized. Ixodidae ticks are responsible for about 10% of human cases of tularemia in most endemic countries, including France and Germany [10]. The mosquito’s role in transmitting tularemia is demonstrated in Scandinavian countries, particularly in Sweden and Finland [8]. Other arthropods such as Tabanidae (notably the deer fly or Chrysops) have also been implicated in transmitting the disease to humans, notably in the United States [8,25,26]. All cited hematophagous arthropods also play a significant role in the transmission of Ft within the animal reservoir. The present review summarizes literature data on the role of mosquitoes in tularemia transmission.

### 2.1. Mosquito

#### 2.1.1. Taxonomy

Mosquitoes are insect arthropods belonging to the order Diptera. Hematophagous mosquitoes are classified in the Nematocera superorder and Culicidae family. This family includes 42 genera and 3563 species differentiated between them by complex morphological criteria. The Culicidae family is divided into two subfamilies: Anophelinae, including three genera; and Culicinae, including 109 genera, divided into 11 tribes [27,28,29]. 

The blood meal necessary for mosquito eggs embryogenesis is obtained by biting humans or animals. During blood meals, females inject a mixture of saliva into their hosts to facilitate their meal. They may also inject pathogens into the host blood capillaries. These pathogens are acquired by the mosquito during previous blood meals and are present in the proboscis (i.e., the elongated appendage from the head) or salivary glands of these arthropods [30].

#### 2.1.2. Geographical Distribution of Arthropod Vector Genera

*Culex (Cx.), Aedes (Ae.),* and *Anopheles (An.)* are the main genera of medical interest distributed around the world, with a different distribution of species according to geographical areas [27,31]. The *Cx. pipiens* and *Cx. quinquefasciatus* species (belonging to the *Cx. pipiens* complex) are the most widely distributed worldwide [32,33]. In recent years, the *Aedes* genus has known a rapid spread of their species in the five continents [34]. The species *Ae. albopictus*, originating from Central Asia, also known as the Asian Tiger Mosquito, is the most invasive species. This species has invaded all continents during the last 20 years, moving from North, Central, and South America to the European continent, affecting the Mediterranean basin, Africa, the Middle East, and Australia [34,35,36]. The species *Ae. japonicus*, *Ae. atropalpus* and *Ae. koriecus* originating from different geographical areas have emerged in many European countries [34,36]. Besides, the species *Ae. aegypti*, known as the yellow fever mosquito, has also expanded in recent years. It has been reported in several European countries, such as Spain, France, Italy, and the Middle East, southern Russia, and North Africa [37]. Finally, the genus *Anopheles* is responsible for transmitting *Plasmodium* to humans, representing the leading cause of death from infectious diseases per year globally. *Anopheles gambiae*, originating from West Africa, is the primary malaria vector in the African Sahel region, Mediterranean areas, and Brazil. In the New World, the species *An. darling* is the most efficient malaria vector [37]. Besides, several species are typically distributed in specific areas according to their climate, such as *An. cinereus cinereus* mainly found in tropical regions of Africa [38]. 

#### 2.1.3. Mosquito-Borne Diseases

Transmission of a pathogen to humans through mosquitoes may occur by two routes: mechanical or biological. Mechanical (passive) transmission implies that the mosquito (called a vector) carries the infectious agent but does not promote its multiplication. The pathogen is present in the mosquito proboscis and inoculated to the host at the time of a blood meal. It can also be present in the abdomen of the mosquito and its droppings deposited on the skin of the host at the time of the blood meal. In this case, infection only occurs if the (human) host scratches the bite site and inoculates the pathogen through their skin, which is usually a reflex action. A biological (active) transmission implies that the pathogen multiplies inside the mosquito (called an intermediate host). It is characterized by the infectious agent’s presence in the mosquito’s salivary glands, where it undergoes multiplication. Transmission and infection of the host automatically occur during a blood meal by injecting the pathogen associated with the mosquito saliva complex. 

Mosquitoes are known to be vectors of many pathogens, including viruses and parasites (Table 1). Malaria caused by different species of the complex *Plasmodium* and transmitted by *Anopheles* mosquito species caused 584,000 deaths in 2013 around the world in 97 countries. Malaria is considered the most virulent human protozoan disease [27,39]. Moreover, viruses are the most dangerous pathogens transmitted by mosquitoes [40]. Mosquitoes’ capacity to transmit bacteria was unknown prior to detecting the species *Rickettsia felis* in *Cx. quinquefasciatus* and the bacteria Fth in several mosquito species [41,42]. Besides, *Bacillus anthracis* has been transmitted experimentally (mechanical transmission) by *Ae. aegypti* and *Ae. taeniorhynchus* mosquito species [43]. However, the mode of transmission of bacteria from the mosquito vector to hosts remains unclear. 

## 3. Mosquitoes and *Francisella*

### 3.1. Human Cases of Tularemia Related to Mosquito Bites

#### 3.1.1. Geographical Areas Concerned: Epidemiology, Climate, Seasonality, and Type of Landscapes

Fth is endemic in Finland and Sweden, where several outbreaks have been reported over the last 30 years [15,54]. In these areas, the disease peaks are recorded between June and September, with more than 50% of cases occurring during August, corresponding to mosquito emerging season [55,56,57]. In these countries, the relationship between mosquito bites and tularemia has been suggested based on epidemiological and clinical data [58,59,60,61]. Between 1931 and 1938, human cases of tularemia in Scandinavia occurred during the year’s warm season, with 80% ulceroglandular forms. Skin inoculation lesions were localized in women’s legs, but arms, face, and neck for men. These localizations are explained by the clothing style, with women wearing short skirts and men wearing trousers. The subjects interviewed could only remember being bitten by mosquitoes [59].

Sweden recorded more than 4792 human tularemia cases between 1984 and 2012, and 4422 between 2000 and 2018 [55,62]. Seven high-risk regions cover 14.2% of the country’s total area and 9.3% of the Swedish population [57]. Epidemiological studies suggest that the high number of tularemia cases during the year’s warm season is positively related to mosquito activity [60]. More recently, between July and September 2019, the city of Gävelborg in central Sweden registered the largest tularemia outbreak recorded in the country in the last 50 years, with 979 cases of which 734 were laboratory confirmed. According to questioned subjects and clinical investigations, infections were related to mosquito bites in 73% of the patients [62]. Moreover, the joint testimonies of affected individuals confirmed having been present at a golf competition, which took place during this period. A survey of mosquitoes collected from puddles on the golf course confirmed the presence of type B *Francisella* in mosquito species collected [62].

In Finland, 5086 tularemia cases were confirmed over 18 years (from 1995 to 2019). All cases occurred between June and October [54]. Five epidemics have been reported over ten years (2000–2010). The largest epidemic occurred in 2000, with 926 tularemia notified cases corresponding to an incidence of 18/100,000 inhabitants [54,56]. During this outbreak, 74% of infected patients suffered from the ulceroglandular form of tularemia. The predominance of this clinical form was compatible with the bacteria’s transcutaneous transmission. Statistical investigations confirmed that mosquitoes were the most probable vector linked to this outbreak [54].

*F. tularensis* is characterized by its complex ecology (two transmission cycles have been suggested, terrestrial and aquatic) and its long-term persistence in several natural environments [9,17,63]. The high-risk areas in Sweden are located in the central and northern parts of the country (Figure 1) [57]. In Finland, high-risk areas are located in the country’s central region and the Gulf of Bothnia (North, Central, and South) (Figure 1) [54]. These areas are characterized by a wet spring, mild summer and autumn, freshwater bodies (lakes and rivers), boreal forests, alpine areas, and different altitudes [55,64]. These conditions are favorable for mosquito species development and prolong the interaction between mosquitoes and tularemia reservoir hosts.

#### 3.1.2. Gender and Age Influence

Tularemia cases affected people with a median age of 45 years (range 0–96 years) in Finland between 1995 and 2013, and people with an age range of 55–69 years in Sweden between 1984 and 2012 [55,56]. In both geographical areas, men were slightly more affected than women (55% of cases) [55,62]. This almost equal distribution between the two sexes is consistent with similar exposure to mosquito bites and contrasts with male predominance observed in most other countries [65]. 

#### 3.1.3. Clinical Manifestations

The majority of tularemia cases reported in Sweden and Finland correspond to skin lesions in the legs combined with inguinal lymphadenopathy, or lesions in arms, face, or neck with axillary or cervical lymphadenopathy. Fever and body aches are also observed. These clinical manifestations correspond to the ulceroglandular form of tularemia associated with mosquito bites [10,66,67]. However, in the rest of Europe, mosquitoes are not recognized to be a vector of tularemia.

Symptoms usually appear after 3 to 5 days incubation, up to 3 weeks [10]. Due to doctors’ extensive experience with tularemia, in Scandinavia, diagnosis of the disease’s ulceroglandular form linked to mosquito bites is usually suspected early. Therefore, serological diagnostic confirmation of tularemia is generally obtained during treatment. Early diagnostic confirmation can be obtained by culture or PCR testing of skin lesions, but these tests are rarely done [68]. The treatment is based on administering an aminoglycoside (streptomycin or gentamycin) for severe diseases. Doxycycline or a fluoroquinolone (especially ciprofloxacin or levofloxacin) are used for mild infections. No resistance to these antibiotics has been detected so far in natural strains of Ft [10].

### 3.2. Mosquito Species Associated with Tularemia 

#### 3.2.1. Major Mosquito Species Associated with Tularemia by Geographic Area

In Sweden, Fth has been naturally detected in twelve different mosquito species: *Ae. punctor, Ae. cinereus, Ae. vexan, Ae. sticticus, Ae. annulipes, Ae. intrudens, Ae. leucomelas, Ae. cantans, An. claviger, An. Maculipennis, Coquillettidia richiardii*, and *Cx. pipiens/torrentium* [41,62,69].

The detection of Fth in several mosquito species reveals a broad spectrum of potential vectors. In Sweden and Finland, *Ae. cinereus* is considered the primary vector of Fth [70]. This vector’s role in transmitting tularemia was demonstrated during the 2019 epidemic in Gävleborg (Sweden) [62]. One of eight pools containing 103 specimens of *Ae. cinereus* collected during the survey following the epidemic tested positive for Fth [62]. However, other endemic mosquito species in Sweden and Finland areas are suspected to be potential vectors of subspecies *holarctica* such *Ae. vexan* and *Ae. sticticus*. These floodwater mosquitoes are known to be nuisance species for humans when present in large numbers. Their larval stages are predators of aquatic amoebae. Ft resists the phagocytosis and digestion by these protozoa, which might promote its persistence in the aquatic environment. Therefore, mosquito larvae may acquire Ft after feeding on infected amoebae. However, *Ft* has never been detected within amoebae in natural surface waters [8,69,71]. 

#### 3.2.2. Particularity of Mosquito Species Associated with Tularemia 

*Ae.* cinereus belongs to the subgenus *Aedes* (Meigen, 1818), for which no species complex or subspecies exist. However, *Ae. cinereus* and *Ae. geminus* are considered twin species, and only the evaluation of the genitalia of the male imago allows accurate differentiation of these species [72]. As a result, many records of *Ae. geminus* are erroneously reported as *Ae. cinereus* [73]. This species is distributed in different bioclimatic floors, such as Russia, Europe, Central Asia, Australia, and North America, revealing their strong adaptation potential [74]. *Ae. cinereus* is a floodwater species characterized by a specific biotope for its development [74]. This mosquito prefers forests and small bodies of stagnant water with a pH ranging from 6 to 9 [73,74,75]. *Ae. cinereus* eggs are highly resistant to external climatic conditions and hatch after the year’s cold period between April and June [76]. The populations of *Ae. cinereus* peak in July and then decrease in September [77]. Females have double activity, nocturnal and diurnal, and are mainly attracted by humans, although they can also bite rodents and birds [74]. 

Floodwater mosquitoes *Ae.* (*Aedimorphus*) *vexans* and *Ae.* (*Ochlerotatus*) *sticticus* are also known to be a nuisance when abundant in a specific area [78]. The geographic extension of these mosquito species is related to precipitation and temperature changes related to global warming [71]. *Ae. sticticus* and *Ae. vexans* are attracted by large rivers with adjacent lowlands that regularly flood [79,80,81,82]. *Ae. vexans* is characterized by migrating long distances up to 48 km, and its robust eggs able to survive in the soil for several years [78]. *Ae sticticus* was also found in new habitats not connected to rivers or lakes, such as abandoned farmland that gets flooded, which increases their abundance and nuisance [71].

### 3.3. Mosquito Life Cycle 

#### 3.3.1. Possible Modes of Francisella tularensis Contamination for Mosquitoes

A competent vector is defined by its ability to acquire and ensure the survival of a pathogen. Mosquitoes are characterized by an aquatic life cycle for the early stages up to the emergence of the imago stage. Mosquito eggs are laid on the surface of the water. The first larval stage (L1) develops inside the egg. Hatching occurs 24 to 48 h after oviposition and gives the 2nd larval development stage (L2). L2 grows to L3 and then to L4. Moults separate the evolution of larval stages. Finally, the L4 stage molt to nymphs, which evolve after 1 to 6 days depending on species and climatic conditions to the imago stage. In the mosquito life cycle, only larvae (L2, L3, and L4) feed on water microorganisms.

There are two possible routes of contamination for mosquitoes. Firstly, the horizontal route of contamination consists of contracting the pathogen from a natural or animal environment. The vertical route consists of transmitting the pathogen through the different development stages of mosquitoes, either from larva to pupa to adult (which is called transstadial transmission) or from adult to offspring (Transgenital or transovarial transmission) [83,84]. Mosquitoes are considered vectors of Fth [58], but these organisms’ interactions are complex and poorly defined. Studies have been carried out in vitro to clarify such interactions, targeting both larvae and adult stages of mosquitoes. Mahajan et al. showed in-vitro that larvae of *Cx. quinquefasciatus* are infected by feeding on Ft (LSV) in both planktonic and biofilm forms, and that the bacterium persists in the larval organism. However, the bacterium was not detected in the pupae after metamorphosis. Moreover, this study localized the bacterium in the midgut and Malpighian cells (situate in Malpighian tubules representing renal excretory tissues) of larvae, suggesting the possible migration of the bacteria from the digestive system to colonize other organs [85]. Further experiments were carried out on *An. gambiae* and *Ae. aegypti* to evaluate Ftn acquisition (strain U112) by mosquito larvae. Results revealed the bacterium’s acquisition by larvae from water, and bacteria remained present after 72 h in larvae of both *An. gambiae* and *Ae. aegypti*. In contrast, a tiny portion (not significant, according to the author) of *An. gambiae* pupae tested after molting were positive for Ftn. After metamorphosis, all adults tested negative for the bacterium. This study also showed that imago can contract the bacterium during a blood meal contaminated by Ftn and that the bacterial load begins to decrease after 72 h in the organism of adult mosquitoes [14].

In 2011, a study confirmed the existence of transstadial transmission of Ft from larvae to adult mosquitoes. After collecting larvae and letting them emerge in the water of their natural environment, Fth was detected in several adult mosquito species [69]. A study conducted by the same team in 2014 confirmed the transstadial transmission of the bacterium acquired in the water by the larval stage to the adult stage of mosquitoes, with 25% of the adults infected. Another study revealed a high prevalence of Ft in mosquitoes at all stages after ingestion of the bacterium by the larvae, suggesting that *Ae aegypti* can maintain the bacterium [41]. These results were confirmed by another similar study, for which 24% of mosquito adults were positive for Fth [86]. All these studies confirmed transstadial transmission of the bacterium from water through larvae to pupal to adult stages. However, another possible route of contamination has been described by Kenny et al. [87]. 

Adult mosquitoes tend to feed on flower nectar. If they are carriers of Ft, they could contaminate flower nectar with this bacteria while feeding. This contaminated nectar could then constitute a temporary Ft reservoir and an active contamination source for other uninfected mosquitoes [87]. Figure 2 presents different cycles of mosquito contamination with Ft and transmission to humans. Figure 3 shows the possible Ft localizations inside mosquitoes and corresponding modes of transmission to humans.

#### 3.3.2. Possible Modes of Transmission of Francisella tularensis to Humans

As for tularemia transmission (Figure 2 and Figure 3), a pioneer study in 1932 suggested mechanical transmission when mosquitoes carrying Ft are crushed on the host’s skin during a blood meal [58]. However, this study remains incomplete due to the lack of information on mosquito species and Ft subspecies [58]. Later studies suggested that female mosquitoes can transmit the bacterium to mammals by biting during a blood meal. This mode of transmission was first shown by experiments using adult mosquitoes (*Ae. Aegypti*) infected in the laboratory during their larval stage with Fth strain 849 (FSC 849). The larvae were let to develop to the adult stage, then placed within small vials containing a mixture of different animals’ blood and covered with parafilm. The adult mosquitoes fed on this artificial source of blood for 48 h. Ft was detected in adult mosquitoes by PCR. Blood samples were tested using real-time PCR and direct fluorescence microscopy. Fth was detected in approximately 20% of the vials. However, it was not possible to cultivate the bacteria from PCR-positive blood meals. 

In the same study, direct transmission of *Francisella* to mice through the bite of an adult mosquito infected at the larval stage by Fth (FSC 849) did not occur. In contrast, mosquito transmission of Ft between diseased and naïve hosts was confirmed experimentally. Mice were infected with Fth (FSC 849), anesthetized, and placed in a container to serve as a food source for uninfected batches of adult female *Ae. aegypti* mosquitoes. Four days later, mosquitoes were placed in a new container with naïve mice. Ft’s transmission was not observed upon the first blood meal of infected mosquito on naïve mice but occurred during the second blood meal. Such observation denotes Ft’s complex transmission cycle inside the mosquito’s organism [41,69]. 

Another study suggested a new transmission cycle [86]. The bacterium was considered associated with the mosquito in a passive resting state (without replication inside the mosquito’s organism) and reanimated upon contact with a mammalian host. Such transmission cycle was hypothesized based on experiments using homogenate of adult mosquitoes (*Ae. aegypti*), infected at the larval stage by Fth (FSC 200) to infect mice intraperitoneally. Mice were monitored for clinical signs of disease for 24 days. Some of them developed clinical symptoms suggestive of tularemia within five days. Spleen homogenates from these animals were found positive by real-time PCR confirming Ft infection [86]. 

Currently, the exact mode (biological or mechanical) of Fth transmission to humans by mosquitoes remains uncharacterized. In contrast, ticks are confirmed vectors of tularemia. Moreover, they are considered the only biological vector of tularemia among arthropods described as Ft vectors [88]. In experimental studies evaluating mosquito transmission of Ft, the species *Aedes aegypti* was most frequently used. It would be of interest to assess other mosquito species’ ability to transmit Ft. This would allow better prediction of the risk of mosquito-borne tularemia in specific geographic areas according to the endemic mosquito species.

## 4. Perspectives

### 4.1. Gaps to Confirm the Role of Mosquitoes as Vectors of Francisella tularensis 

An inventory of clinical and epidemiological data linking tularemia to mosquito bites has led to the suspicion of these arthropods’ role as vectors of Ft. Available data mainly suggest mechanical transmission of the bacterium by mosquitoes. Still, many elements are missing to confirm this relationship, including responses to the following questions:-Does Fth multiply in the salivary glands of mosquitoes?-After ingestion, how does Fth resist the digestive enzymes of mosquitoes? Can it multiply in the mosquito’s digestive system and travel up to colonize the salivary glands?-What is the preferred microhabitat of Fth in mosquitoes?-What is the involvement of amoebae in the transmission of Fth to larval stages of mosquitoes?-What are the mechanisms used by Fth to survive through the various molts during the development of mosquito larval stages to adult?-Recently, phylogeography studies have shown several subpopulations of Fth around the word. In Scandinavia, Fth subpopulations B.Br.013/014, B.Br.012/013, B.Br.007/008, and B.Br.OSU18 are the most predominant [89]. Are there specific relationships between Fth subpopulations and mosquito species in Scandinavian areas?

Since tularemia’s epidemiology and ecology depend on a geographical context, what is the biotope factor that causes mosquitoes to act as vectors of the bacterium in specific areas in Sweden and Finland?

-What are the particularities of the mosquito species capable of transmitting Fth?-Are there mosquito species more adapted to transmit Fth than others?

### 4.2. Why Transmission of Tularemia by Mosquitoes is Restricted to Finland and Sweden?

Extensive epidemiological investigations have confirmed the role of mosquitoes in tularemia transmission in Sweden and Finland. This mode of transmission might exist in other parts of the world where extensive public health investigations have not been performed. However, mosquito-borne tularemia cases are unlikely to be frequent in geographic areas where these infections’ clinical and epidemiological features are not or rarely observed. This evidence leads us to a three-parameter equation: geographical areas (Finland and Sweden), mosquitoes, and Fth. Several hypotheses can be drawn:-Climate and geology of Finland and Sweden could be factors favoring the association of Fth with mosquitoes?-Native mosquito species in Finland and Sweden might have undergone adaptations to the Fth, creating new vector-bacterium relationships

### 4.3. Transovarian Transmission

The existence of transovarial transmission of Fth from females mosquitoes to their offspring has not been proven so far. However, a study suggested that the fecundity of adults from larvae exposed to Ft is reduced, which relates to this bacterium’s presence in the reproductive organs’ tissues [85]. Two hypotheses for transovarial transmission of Fth can be made, knowing that Ft has an intracellular lifecycle: 1/the presence of the bacterium in the tissues of reproductive organs of female mosquitoes may affect the eggs during their embryonic development; 2/the presence of the bacterium in the sperm of male mosquitoes may affect female mosquito eggs during fertilization. Moreover, Lundstrom et al. found mosquitoes PCR-positive for Fth, whereas the water they were in was PCR negative [58]. This result justifies investigating a possible transovarial transmission of Fth.

### 4.4. Mosquito Control and Tularemia

Due to their public health risks, different control strategies have been developed against mosquito populations over time. Old methods consist of the use of oil or larvivorous fish as larvicides [90]. In the industrial era, chemical substances were developed to be used as larvicide and adulticide. A few years later, these substances were banned due to their negative impacts on the environment and humans and animal health [91,92]. Moreover, chemical substances, such as Pyrethroids, caused multiple insecticide resistance mechanisms in mosquito vector species [93,94]. These limitations turned researchers’ attention to biological control methods as new safe and efficient alternatives, such as fungi, bacteria, gene drive, and sterile male technologies against mosquito populations. 

Mosquitoes population control is an important step in the limitation of transmissible infection propagation. By eliminating or diminishing mosquito larval and adult stages, the pathogens transmission cycle can be broken. In Ft’s case, decreasing the larval and adult stages is necessary, as these two stages can acquire this bacterium, and the adults can then transmit Ft to humans. [14,85]. It will be important to intensify the mosquito control campaign during the transitional periods between the hot and cold seasons and during the hot season. The frequency of rains at the beginning and end of the cold season creates larval shelters. 

## 5. Conclusions

This review highlights the relationship between mosquitoes and tularemia transmission based on an extensive literature inventory. Our bibliographic search confirms mosquitoes’ involvement in Fth transmission in Sweden and Finland, causing hundreds of tularemia cases each year in these countries. There is a need to continue investigating these arthropods’ potential role in other geographic areas. Besides, the continuous spread of different mosquito species and global warming could favor the emergence of the mosquito-Ft cycle in new geographical regions. Many ways of mosquito contamination by Fth may be considered, while the mechanisms of transmission of this bacterium from mosquitoes to humans are still unclear. Further research is needed to characterize the natural cycle of mosquito contamination with Ft and understand these arthropods’ role as passive or active vectors for tularemia transmission to humans. 

## Figures and Tables

**Figure 1 microorganisms-09-00026-f001:**
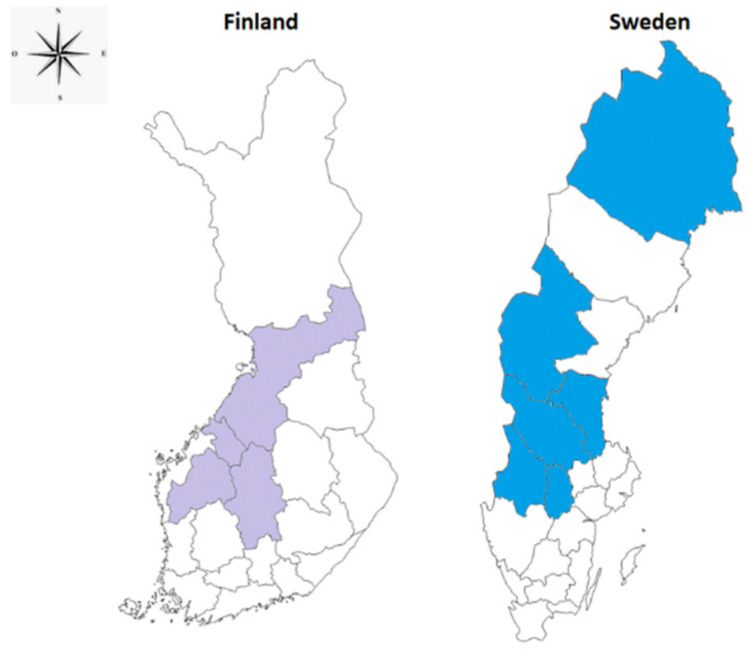
Tularemia high-risk areas in Finland highlighted with purple color and in Sweden highlighted with blue color [54,57].

**Figure 2 microorganisms-09-00026-f002:**
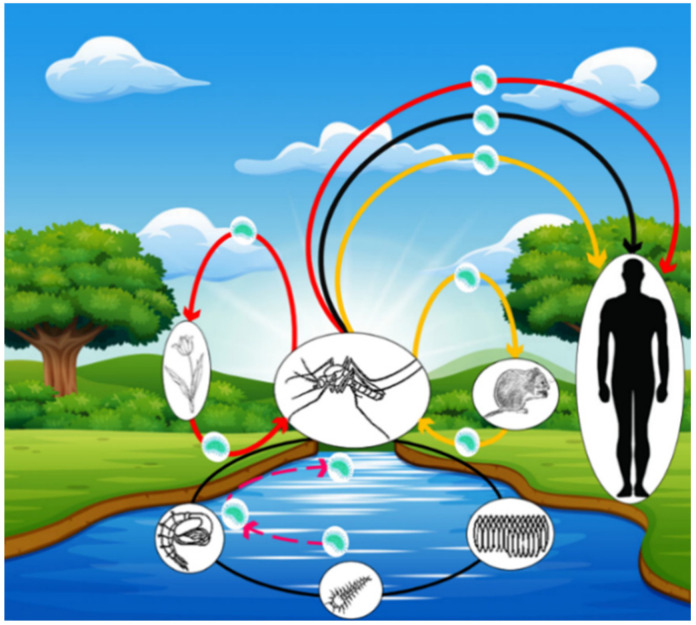
Cycle of *F. tularensis* transmission through mosquitoes. The yellow cycle illustrates mosquito contamination through a terrestrial animal reservoir (especially rodents); the Black cycle implies the acquisition of *F. tularensis* by the larval stage of mosquitoes and transstadial transmission to the adult stage (Pink dashed line), which subsequently becomes infectious. The red cycle illustrates an unconfirmed pathway of mosquito contamination through *F. tularensis*-contaminated flower nectar.

**Figure 3 microorganisms-09-00026-f003:**
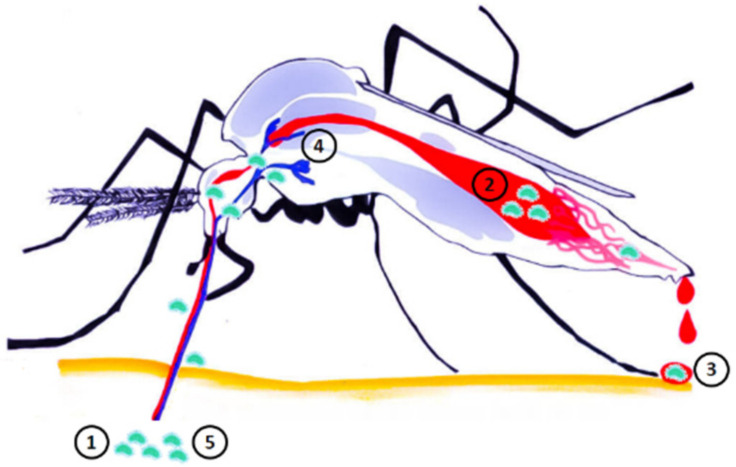
*F. tularensis* possible localizations inside mosquitoes and corresponding modes of transmission to humans. Mechanical contamination cycle via mosquito dejection: *F. tularensis* is ingested during a blood meal in an infected host ①, the bacterium ingested is located in the mosquito midgut ②, *Francisella tularensis* is then digested, crosses the Malpighian tube to be rejected onto a new host dermis during a blood meal ③. Mechanical contamination cycle via mosquito saliva: *F. tularensis* is ingested during a blood meal in an infected host ①, the bacterium stays hooked inside mosquito’s proboscis (i.e., the elongated appendage from the head), and then is reinjected to a new host via the contaminated saliva during a new blood meal ⑤. Biological contamination cycle: *F. tularensis* is ingested during a blood meal in an infected host ①, the bacterium migrates to the salivary glands and undergoes several multiplications ④, the bacterium is then injected into a new host via the contaminated saliva during a new blood meal ⑤.

**Table 1 microorganisms-09-00026-t001:** Major mosquito-borne diseases.

	Human Pathogens	Vectors	References
**Viruses**	Dengue Fever virus	*Aedes*	[44]
	Zika virus	*Aedes*	[45]
	West Nile virus	*Culex*	[46]
	Rift Valley Fever virus	*Aedes and Culex*	[47]
	Chikungunya virus	*Aedes*	[48]
	Yellow Fever virus	*Aedes*	[44]
	Japanese Encephalitis virus	*Culex*	[49]
	Saint Louis Encephalitis virus	*Culex*	[50]
**Parasites**	*Wuchereria boncrofti*	*Aedes, Culex, and Anopheles*	[51]
	*Brugia Malaya*	*Aedes, Culex, and Anopheles*	[51]
	*Dirofilaria immitis*	*Aedes, Culex, and Anopheles*	[52]
	*Dirofilaria repens*	*Aedes, Culex, and Anopheles*	[52]
	*Plasmodium knowlesi*	*Anopheles*	[53]
	*Plasmodium ovale*	*Anopheles*	[53]
	*Plasmodium vivax*	*Anopheles*	[53]
	*Plasmodium falciparum*	*Anopheles*	[53]
	*Plasmodium malariae*	*Anopheles*	[53]
**Bacteria**	*Bacillus anthracis*	*Aedes* and *Culex*	[43]
	*Francisella tularensis*	*Aedes* and *Culex*	[41]
	*Rickettsia felis*	*Cx. quinquefasciatus*	[42]

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
