# Peer review of "Tularemia as a Mosquito-Borne Disease"

_microorganisms, 2020, doi:10.3390/microorganisms9010026_

Round 1

Reviewer 1 Report

The review article by Abdellahoum, Maurin, and Bitam discusses the interaction between mosquitoes and Francisella tularensis. This is an area of tularemia that has been severely overlooked and deserves additional attention. The manuscript does a fair job at providing background information on tularemia and mosquitoes, but it is difficult to understand the relevance of the information provided at times. The manuscript would benefit immensely from the authors providing more context and interpretation for the information in the review so the readers understand why studies that are mentioned are important. I offer potential areas for improvements and additional editorial comments for improvement of what is generally an interesting article.   Content Comments: Line 51: It would be appropriate to refer to Ft as a “facultative” intracellular bacteria. Line 68 Type A and Type B distribution is mentioned in an earlier section (section 1.2; lines 51-56). It would be beneficial to combine section 1.2 and 1.3 so the reader explicitly makes the connection as to what subspecies is typically found in each specific area. Otherwise section 1.3 is redundant. Lines 141-142: The statement “association between mosquitoes and bacteria are still not confirmed” contradicts many findings provided in this manuscript of bacteria recovered from mosquitoes and links between the insect and tularemia. I am not sure what the authors are trying to say with this comment. Line 141: Is there a reference for the link between mosquitoes and B. anthracis? Line 198: Is some other vector suspected or the field studies have not performed for the rest of Europe? Line 209. The different mosquito species may be better represented in a table form with, if possible, the relative geographical range for each of these. This would help the reader understand if some are only of the species are only found near or around Nordic countries such as Sweden or if they have a broader range. This might also be helpful for later discussion such as Section 3.2.2 (Line 225-227). Line 246: As someone who is not well versed in mosquito biology, I was not sure what the terms L2, L3, and L4 corresponds to for the lifecycle. An explanation (or a figure) here would be beneficial for this review. Line 260. An interesting area to explore for discussion that is hinted in this manuscript would be if Ft enters the vector through water/environment or if Fth is passed via biting an infected host. Figure 1 is useful in understanding the mosquito life cycle, and different routes of transmission, though it would be helpful to include information on the amount of time spent in each stage of life/ how long between blood meals/ how long adults typically survive/ etc. Would be helpful to describe what the “Malpighian tube” is or located in a mosquito. Lines 321-323: I am assuming the authors are referring to the potential of the bacteria being in the viable but not culturable state but they never use that term. Is there a reason not to? Lines 348-355: Is the reason for the link between mosquitoes and Ft in Finland and Sweden actually due to some environmental aspect of that the public health communities have performed much more rigorous testing versus other countries with tularemia in the northern hemisphere? ​   Is it worth discussing/ mentioning the possibility of more mosquitoes due to climate change and more spread/ carriage of Ft? Since there are major gaps in this field, and few studies have looked at transmission of Ft from mosquitoes, it would be helpful to describe these few studies in more detail and outline the models that were used- which Ft strains? which species of mosquito? How was transmission assessed? Several of these studies also looked at presence of the bacteria in mosquitoes isolated from the environment, while others infected mosquitoes in a laboratory and tested ability to transmit the bacteria. Specifically, lines 317-324 glosses over these studies and should be expanded to address this information.. Along these lines, it would be helpful to comment on which models are successful, or does a good model for transmission not exist yet and needs to be refined/ developed to answer these questions? Which mosquito species would be the most relevant to test in these models? Editorial Comments: Lines 24, 284 : Perhaps a better word choice would be “infection” versus “contamination”? Line 46: Does “Tularemia” need to be capitalized? Line 196: Awkward sentence and not sure what the authors mean. “Besides, fever and body aches were observed”. Line 272: “mosquito” adults Line 276: ....with this bacteria “while feeding”. Line 295: The #2 legend within the figure was difficult to recognize. Line 312: Should be “bite”.

Author Response

Responses to reviewers’ comments

Reveiwer # 1

Comments and Suggestions for Authors

The review article by Abdellahoum, Maurin, and Bitam discusses the interaction between mosquitoes and Francisella tularensis. This is an area of tularemia that has been severely overlooked and deserves additional attention. The manuscript does a fair job at providing background information on tularemia and mosquitoes, but it is difficult to understand the relevance of the information provided at times.

The manuscript would benefit immensely from the authors providing more context and interpretation for the information in the review so the readers understand why studies that are mentioned are important.

I offer potential areas for improvements and additional editorial comments for improvement of what is generally an interesting article.

 Content Comments:

Line 51: It would be appropriate to refer to Ft as a “facultative” intracellular bacteria.

Line 50: the term “facultative” was added

Line 68 Type A and Type B distribution is mentioned in an earlier section (section 1.2; lines 51-56). It would be beneficial to combine section 1.2 and 1.3 so the reader explicitly makes the connection as to what subspecies is typically found in each specific area. Otherwise section 1.3 is redundant.

Lines 71-81. All information about tularemia geographical distribution has been placed in part 1.3 

Lines 141-142: The statement “association between mosquitoes and bacteria are still not confirmed” contradicts many findings provided in this manuscript of bacteria recovered from mosquitoes and links between the insect and tularemia. I am not sure what the authors are trying to say with this comment.

Lines 156-157: We agree with the reviewer and we have changed the sentence accordingly.

Line 141: Is there a reference for the link between mosquitoes and B. anthracis?

Lines 155-156: Yes, in reference number 43. We added a short sentence on the link between mosquitoes and B. anthracis.

Line 198: Is some other vector suspected or the field studies have not performed for the rest of Europe?

Yes, there are other vectors. Ixodidae ticks are confirmed vectors of tularemia in almost all tularemia endemic areas (lines 364-366). However, no extensive evaluation of the role of mosquitoes in the transmission of tularemia have been done (or published) in Europe outside Sweden and Finland. This is likely due to lack of epidemiological evidence linking tularemia to mosquito bites. We focused our review on the interactions between mosquitoes and tularemia, and thus we did not describe other known or potential arthropod vectors of thi disease.

Line 209. The different mosquito species may be better represented in a table form with, if possible, the relative geographical range for each of these. This would help the reader understand if some of the species are only found near or around Nordic countries such as Sweden or if they have a broader range. This might also be helpful for later discussion such as Section 3.2.2 (Line 225-227).

It would be difficult to build a table showing the relative geographical range of mosquito species. The involved mosquito genera are present in all Europe and even in other continents. The distribution of each mosquito species is not limited to a specific country or area due to the globalization and the exchanges between the continents. Nowadays, many mosquito species have colonized new areas and shown great potential of adaptation to the new biotopes in which they emerged. This is the case, for example, of Aedes albopictus, which spread from central Asia to all continents and areas with different bioclimatic conditions.

Line 246: As someone who is not well versed in mosquito biology, I was not sure what the terms L2, L3, and L4 corresponds to for the lifecycle. An explanation (or a figure) here would be beneficial for this review.

Lines 270-275: we added a paragraph explaining mosquito lifecycle. 

Line 260. An interesting area to explore for discussion that is hinted in this manuscript would be if Ft enters the vector through water/environment or if Fth is passed via biting an infected host.

Section 3.3.1, explains the different ways of vector contamination with Fth. Mosquitoes can be infected with Ft from water contaminated with this bacterium in the larval stage. They can acquire Ft at the adult stage by biting an infected host and possible by feeding on contaminated flower nectar.

Figure 1 is useful in understanding the mosquito life cycle, and different routes of transmission, though it would be helpful to include information on the amount of time spent in each stage of life/ how long between blood meals/ how long adults typically survive/ etc. Would be helpful to describe what the “Malpighian tube” is or located in a mosquito.

In this figure, we wanted to illustrate the different modes of acquisition and transmission of Ft by mosquitoes. We think that adding the above information would make reading the figure too complex. The main steps of the mosquito lifecycle are described lines 270-275. 

Lines 321-323: I am assuming the authors are referring to the potential of the bacteria being in the viable but not culturable state but they never use that term. Is there a reason not to?

In this section we highlighted the state of Ft inside mosquitoes. For the mechanical cycle, the bacteria are considered to remain viable but in quiescent state, without multiplication. However, these bacteria usually can still be isolated in culture from these arthropods. Therefore, the term viable but not cultivable seems not adapted. The multiplication of bacteria inside mosquitoes correspond to the biological cycle. We added a paragraph lines 358 to 363 to further explain results of the corresponding studies.  

Lines 348-355: Is the reason for the link between mosquitoes and Ft in Finland and Sweden actually due to some environmental aspect of that the public health communities have performed much more rigorous testing versus other countries with tularemia in the northern hemisphere? ​

Sweden and Finland have been recognized as tularemia endemic areas since the beginning of the 20th century. Clinicians and scientists noticed a relationship between mosquito bites during the summer and ulceroglandular forms of tularemia occurring during the same season. Skin lesions developed at the site of mosquito bites, as reported by affected people. Epidemiological studies have then rigorously confirmed the role of mosquitoes as vectors of Ft in these countries. This mode of transmission might exist in other parts of the world where extensive public health investigations have not been performed. However, tick-borne tularemia cases are unlikely to be frequent in geographic areas where the clinical and epidemiological features of these infections are not or rarely observed. We have added a paragraph in the discussion section (lines 411 to 415).

Is it worth discussing/ mentioning the possibility of more mosquitoes due to climate change and more spread/ carriage of Ft?

We have added sentence in the conclusion, lines 455-456.

Since there are major gaps in this field, and few studies have looked at transmission of Ft from mosquitoes, it would be helpful to describe these few studies in more detail and outline the models that were used- which Ft strains? which species of mosquito? How was transmission assessed? Several of these studies also looked at presence of the bacteria in mosquitoes isolated from the environment, while others infected mosquitoes in a laboratory and tested ability to transmit the bacteria.

Specifically, lines 317-324 glosses over these studies and should be expanded to address this information. Along these lines, it would be helpful to comment on which models are successful, or does a good model for transmission not exist yet and needs to be refined/ developed to answer these questions? Which mosquito species would be the most relevant to test in these models?

All of this information has been added in section 3.3.2 (lines 336-370)

Editorial Comments:

Lines 24, 284: Perhaps a better word choice would be “infection” versus “contamination”?

We agree. The term “infection” is better adapted for animals, including arthropods

Line 46: Does “Tularemia” need to be capitalized?

No, we have changed the word to “tularemia”.

Line 196: Awkward sentence and not sure what the authors mean. “Besides, fever and body aches were observed”.

Sentence modified line 214

Line 272: “mosquito” adults

Done

Line 276: ....with this bacteria “while feeding”.

Done

Line 295: The #2 legend within the figure was difficult to recognize.

We put the legend inside the digestive system to show the way that bacteria progress. 

It could be placed outside the digestive system for better visibility if necessary.

Line 312: Should be “bite”.

Yes.

Reviewer 2 Report

The manuscript (Manuscript ID: microorganisms-1041865) entitled " Tularemia as a mosquito-borne disease" is comprehensive and well written.

One minor comment, in section 1.2, lines 59-65, please address shortly the lack of confirmed vaccine for FT, specifically in the context of the reason for categorization as a class A biological agent. 

Author Response

The manuscript (Manuscript ID: microorganisms-1041865) entitled " Tularemia as a mosquito-borne disease" is comprehensive and well written.

One minor comment, in section 1.2, lines 59-65, please address shortly the lack of confirmed vaccine for FT, specifically in the context of the reason for categorization as a class A biological agent. 

Paragraph added lines 58-63.

Reviewer 3 Report

This is a well written and informative review. Editing of the language and sentence structure would be useful in some places. I just have a few comments:

Line 21: This makes it sound like the other 2 subspecies don’t cause human disease at all.

Lines 109-110: Please re-word this sentence.

Line 140: Change ‘since’ to ‘prior to’.

Line 135-142: In this section, particularly when discussing bacteria, it would be good to talk about the different ways in which mosquitoes can act as vectors, i.e. mechanical transmission vs. biological transmission. This is especially important because of the article's focus on F. tularensis, but also because Bacillus anthracis is included in table 1. From my understanding mosquitoes have only been found to be mechanic vectors of this pathogen, not true vectors of. I know you do discuss this later on (lines 305 onward), but I think it needs to be discussed earlier in the text to set the context.

Table 1: It would be good to provide references for these pathogens.

Line 179-183: It would be helpful to have a map of this region.

Lines 317-324: It would be good to compare what is discussed here to what is known about the role of ticks in F. tularensis transmission.

Line 335: One of the questions is about the involvement of amoebae in the transmission of F. tularensis between different life stages in mosquitoes, but the only mention of this in the text is a single sentence (lines 219-220). It would be helpful to provide a bit more information on this.

Lines 354-355: Don’t the mosquito species suspected of being F. tularensis vectors have wider geographic distributions than just Finland and Sweden? If you are suggesting that the local mosquito populations have evolved to host F. tularensis, what evidence exist to suggest these local populations are genetically different to those found elsewhere?

Lines 372-374: Also other methods such as gene drive and sterile male technologies.

Author Response

Responses to reviewers’ comments

Reveiwer # 3

This is a well written and informative review. Editing of the language and sentence structure would be useful in some places.

English editing has been performed throughout manuscript

I just have a few comments:

Line 21: This makes it sound like the other 2 subspecies don’t cause human disease at all.

Sentence modified

Lines 109-110: Please re-word this sentence.

Sentence re-worded (lines 212-213)

Line 140: Change ‘since’ to ‘prior to’.

Done

Line 135-142: In this section, particularly when discussing bacteria, it would be good to talk about the different ways in which mosquitoes can act as vectors, i.e. mechanical transmission vs. biological transmission. This is especially important because of the article's focus on F. tularensis, but also because Bacillus anthracis is included in table 1. From my understanding mosquitoes have only been found to be mechanic vectors of this pathogen, not true vectors of. I know you do discuss this later on (lines 305 onward), but I think it needs to be discussed earlier in the text to set the context.

Done, we moved the part explaining different ways of pathogens transmission by mosquito from paragraph 3.3.2 to paragraph 2.1.3.

Table 1: It would be good to provide references for these pathogens.

Done

Line 179-183: It would be helpful to have a map of this region.

Done, page 6

Lines 317-324: It would be good to compare what is discussed here to what is known about the role of ticks in F. tularensis transmission.

Done, lines 365-367

Line 335: One of the questions is about the involvement of amoebae in the transmission of F. tularensis between different life stages in mosquitoes, but the only mention of this in the text is a single sentence (lines 219-220). It would be helpful to provide a bit more information on this.

Done, lines 239-243

Lines 354-355: Don’t the mosquito species suspected of being F. tularensis vectors have wider geographic distributions than just Finland and Sweden? If you are suggesting that the local mosquito populations have evolved to host F. tularensis, what evidence exist to suggest these local populations are genetically different to those found elsewhere?

It was only a hypothesis. We changed the sentence (Lines 404-405). There is currently no evidence supporting that local mosquito populations are genetically different

Lines 372-374: Also other methods such as gene drive and sterile male technologies.

Added, lines 423-424

Round 2

Reviewer 1 Report

In general, the manuscript has been approved but several areas of editorial corrections are necessary. Line 47, I would recommend workding "... proposed the name tularemia for the disease... Line 118. Not sure what is meant by the sentence, "Mosquito bites represent their medical interest"? I believe this sentence could be removed. Lines 143-154. It appears from tracked changes that most of this paragraph has been removed? Was that the intent of the authors? If so, the remaining sentence (indicated in blue) is not grammatically correct. Line 164: Not sure what is meant by the sentence "... the bacteria transmission WAY from the mosquito vector..." I believe "way" could be removed to make this sentence correct. Line 285: Is this supposed to be "finally"? Lines 352-354: This is an awkward sentence. Would it be correct to remove "in its organism" from it? Line 356: Should this be "infection" not "contamination"? Lines 357-361: Not sure if this sentence is correct based upon the tracked changes.

Author Response

Line 47, I would recommend workding "... proposed the name tularemia for the disease...

Done

Line 118. Not sure what is meant by the sentence, "Mosquito bites represent their medical interest"? I believe this sentence could be removed.

We agree that this sentence was unclear and we deleted it.

Lines 143-154. It appears from tracked changes that most of this paragraph has been removed? Was that the intent of the authors? If so, the remaining sentence (indicated in blue) is not grammatically correct.

The paragraph in blue was only a response to the reviewer’s comment. In the first revised version, we transferred the paragraph “Transmission of a pathogen to humans through mosquitoes may occur by two routes: mechanical or biological. ……………Transmission and infection of the host automatically occur during a blood meal by injecting the pathogen associated with the mosquito saliva complex.” from section 3.3.2 “Possible modes of transmission of Francisella tularensis to humans “ to section 2.1.3 “Mosquito-borne diseases”. We made this change because the information provided in this paragraph applies to all mosque-borne diseases and not only F . tularensis. No paragraph has been removed in this latter section.

Line 164: Not sure what is meant by the sentence "... the bacteria transmission WAY from the mosquito vector..." I believe "way" could be removed to make this sentence correct.

We have changed the sentence to “However, the mode of transmission of bacteria from the mosquito vector to hosts remains unclear.”

Line 285: Is this supposed to be "finally"?

Yes, the word has been corrected

Lines 352-354: This is an awkward sentence. Would it be correct to remove "in its organism" from it?

We have changed the sentence to “Mechanical (passive) transmission implies that the mosquito (called a vector) carries the infectious agent but does not promote its multiplication.” Please note that this sentence is now in the section 2.1.3 “Mosquito-borne diseases”

Line 356: Should this be "infection" not "contamination"? Lines 357-361: Not sure if this sentence is correct based upon the tracked changes.

Contamination has been replaced by infection. For better clarity, the paragraph has been changed to: “It can also be present in the abdomen of the mosquito and its droppings deposited on the skin of the host at the time of the blood meal. In this case, infection only occurs if the (human) host scratches the bite site and inoculates the pathogen through their skin, which is usually a reflex action.”